

# Evaluation of epidemiological lectures using peer instruction: focusing on the importance of ConcepTests

Toshiharu Mitsuhashi

Center for Innovative Clinical Medicine, Okayama University Hospital, Okayama, Japan

## ABSTRACT

**Background**. In clinical research, the ability to properly analyze data is a necessary skill that cannot be learned simply by listening to lectures. Interactive classes, such as Peer Instruction (PI), are required to help medical students understand the concept of epidemiology for future valid research. In PI lectures, ConcepTests are conducted to confirm and deepen students' understanding of the lecture material. Although it is important to evaluate PI lectures, there have been no studies conducted on PI lectures in epidemiology. This study employed the ConcepTest to evaluate PI lectures in a medical school epidemiology class to measure the efficiency of active learning techniques and the usefulness of ConcepTests in determining effective active learning approaches.

**Methods**. The PI lecture was conducted as part of an existing epidemiology class for fourth-year medical students at Okayama University on October 17, 2019. The lecture was conducted as follows. The lecturer taught the fundamental concepts of epidemiology and presented the ConcepTest to students. After answering the test, students were provided with the answer distribution, followed by peer discussion. After the discussion, students answered the ConcepTest again, and a new answer distribution was presented. Subsequently, the lecturer announced the correct answers and delivered a commentary. The ConcepTest comprised five questions, each related to fundamental concepts of epidemiology. Students' responses to five ConcepTests were collected and analyzed by calculating the proportion of correct answers before and after the discussion, as well as PI efficiency to evaluate the PI lecture.

**Results**. Overall, 121 students attended the epidemiology lecture. The proportion of correct answers before the discussion ranged from 0.217 to 0.458, and after the peer discussion they ranged from 0.178 to 0.767. The PI efficiency ranged from −0.051 to 0.657, and was higher than the theoretical value in three ConcepTests. The efficiency was about the same as the theoretical value in one ConcepTest, and lower than the theoretical value in another.

**Conclusion**. In this study, the efficiency of a PI lecture was determined by calculating the PI efficiency of each ConcepTest. The results showed that the educational efficiency of a ConcepTest in epidemiology lectures can be widely distributed, ranging from efficient to inefficient. Particularly in three ConcepTests, the proportion of correct answers after the discussion and the PI efficiency were higher than the theoretical value. This suggests that PI lectures can be useful in epidemiology education with the efficient use of ConcepTests.

Corresponding author
Toshiharu Mitsuhashi,
mitsuh-t@cc.okayama-u.ac.jp

## INTRODUCTION

Recently, there has been growing interest in data science education within academic programs (*Huppenkothen et al., 2018*; *Guzman et al., 2019*; *Park et al., 2019*). In medical research in particular, clinical data-based research is becoming increasingly important (*Macleod et al., 2014*; *Sandercock & Whiteley, 2018*; *Fridsma, 2018*). To learn how to appropriately analyze clinical data and conduct clinical research, medical students must take data science undergraduate classes (e.g., epidemiology). In other words, high-quality epidemiology education is required for high-quality clinical research.

It has been shown that data science cannot be learned efficiently through lectures alone (*Haller & Krauss, 2002*). Therefore, Peer Instruction (PI) has recently attracted the attention of educational practitioners (*Hilborn, 1997*). PI is a type of interactive learning that is easy to incorporate into conventional lecture styles and can be conducted even with a large number of students (*Crouch, 1998*), unlike small-group interactive methods such as problem-based learning. PI education has been introduced in various fields, and PI lectures have been conducted primarily in the fields of physics, mathematics, and engineering (*Pilzer, 2001*; *Crouch & Mazur, 2001*; *Schmidt, 2011*). The effects of PI lectures on students' conceptual understanding in these fields were shown to improve students' understanding and satisfaction (*Giuliodori, Lujan & DiCarlo, 2006*; *Relling & Giuliodori, 2015*; *University College London, 2018*; *Vázquez-García, 2018*). Although educational methods that enable better understanding of epidemiology concepts have been studied (*Goldmann et al., 2018*; *Dankner et al., 2018*; *Sohn et al., 2019*), limited research has been conducted on PI lectures in epidemiology (*Katyal et al., 2016*). Considering its effective application in other fields, PI may be a promising educational method in epidemiology.

During PI lectures, important concepts that students should learn are tested using the ConcepTest, through which a lecturer can confirm students' understanding, and students can, in turn, deepen their understanding. The effectiveness of the ConcepTest is one of the important factors determining its success or failure (*Crouch & Mazur, 2001*; *Nitta, 2010*). Additionally, students' responses to the ConcepTests provide further insights into student learning. However, there have been no reports of ConcepTest evaluations for PI in epidemiology. That is, it is not clear whether the learning efficiency per ConcepTest in epidemiology is high or low. This knowledge gap needs to be filled. Therefore, this study aims to report the effectiveness of ConcepTests used in PI lectures in epidemiology.

## MATERIALS & METHODS

### Ethical consent

The study was approved by the Okayama University Graduate School of Medicine, Dentistry and Pharmaceutical Sciences, and Okayama University Hospital Ethics Committee (approval number K1909-037). The purpose and methods of the study were adequately presented to the students on paper, informed consent was obtained, and the students were told that they were free to withdraw participation for any reason.

## Study design and settings

The PI lecture in this study was presented on October 17, 2019 to an existing epidemiology class for fourth-year medical students at Okayama University. As PI lectures had not been previously presented at the Okayama University School of Medicine, the PI lecture style was explained to the students. Students' answers to five ConcepTests conducted during the PI lecture were collected. The author designed and oversaw all exercises and ConcepTests.

## Lecture contents

Students were not informed in advance that they would be presenting PI lectures, nor were they instructed to make special preparations for the PI lectures. The PI lecture was conducted as part of an epidemiology and statistics exercise. The lecture time was 80 min, and it was divided into a series of five short sections that dealt with the following topics:

1. Epidemiological indicators (risk, incidence, and prevalence).
2. Descriptive epidemiology (spot map and epidemic curve).
3. Cohort and case-control study (study concept and interpretation of a two-by-two table).
4. Random error (error evaluation and interpretation of confidence interval).
5. Systematic error (selection, information, and confounding biases).

After the PI lecture, statistical analysis was performed using the statistical software Epi Info 7 (http://www.cdc.gov/epiinfo/index.html).

## Mentimeter as a PI tool

In this study, Mentimeter (http://www.mentimeter.com), a web-based interactive presentation software often used in interactive education, was used as a PI tool (*Andriani, Dewi & Sagala, 2019*; *Moorhouse & Kohnke, 2020*). Via the Internet, students can answer questions incorporated into the presentation and their answers are tabulated immediately. The lecturer can present the compiled results to the student at an appropriate time. Using the Mentimeter, the author practiced several times before the PI lecture to ensure that the students' responses were captured for the ConcepTest.

## ConcepTest in the PI lecture

After explaining each of the five themes, a ConcepTest was conducted to confirm students' understanding of the lecture material. The contents of the ConcepTests are shown in the Table A1. ConcepTest #1 presented a question about whether the risk or prevalence could be calculated from the information provided. The epidemic curve presented in ConcepTest #2 concerned a food poisoning incident that occurred on an aircraft in 1984 and was quoted from the literature (*Tauxe et al., 1987*). This test asked students to interpret the epidemic curve. ConcepTest #3 consisted of a two-by-two table about thalidomide teratogenicity and was quoted from a Japanese book titled *Shimin no tameno ekigaku nyūmon* (Introduction to Epidemiology for Citizens) (*Tsuda, 2003*). This test asked students to interpret the two-by-two table for a case-control study. ConcepTest #4 provided point estimates and confidence intervals for risk ratios, which students were asked to interpret. ConcepTest # 5 asked about research methods that are less likely to have selection and/or information

bias. ConcepTests #1, #4, and #5 were generated from hypothetical scenarios. The topic of each ConcepTest corresponds to each of the five topics of the preceding lecture.

## PI method overview

One set of the PI was performed in about 6 min as follows:

  I.   The lecturer presents the ConcepTest after the presentation.
 II.   Students answer via Mentimeter.
III.   The lecturer presents the answer distribution.
 IV.   Students subsequently hold a discussion to convince each other of the correctness of their answers.
  V.   Students answer via Mentimeter again.
 VI.   The lecturer presents the new answer distribution.
VII.   The lecturer announces the correct answer and gives a commentary.

During Step IV, students were instructed to discuss their answers with other students seated nearby. It can be assumed that students seated nearby would have an easy relationship with each other, so this method was adopted. Therefore, in some cases, students with the same opinions held a discussion with each other, while in other cases, students with opposite opinions conducted a discussion.

## Statistical analysis

Simple tabulations were performed using Excel 2019 (version 1911; Microsoft Corporation, Redmond, WA, USA), and graph drawings were created using Stata software (Stata Corporation, version 16.1, College Station, TX, USA).

## Student attributes

The number and gender of respondents for each ConcepTest were tabulated.

## PI efficiency

The effectiveness of the ConcepTest was measured using PI efficiency $\eta$, defined with the help of Hake's standardized gain (*Kaneta & Nitta, 2009*; *Nitta, 2010*), as follows:

$$\eta = \frac{N_a - N_b}{1 - N_b}$$

where the proportion of correct answers before and after the discussion is denoted by $N_b$ and $N_a$, respectively. It is considered that $\eta$ reflects the ease of understanding gained through PI.

## Theoretical value of PI efficiency and proportion of correct answers after the discussion

The theoretical value of $N_a$ and $\eta$ can be expressed as a function of $N_b$, with some assumptions (*Nitta, 2010*), as follows:

$$N_a = 2N_b - N_b^2$$

$$\eta = N_b$$

The theoretical value was calculated according to these formulas, and the difference from the measured value is shown in Figs. 1 and 2.
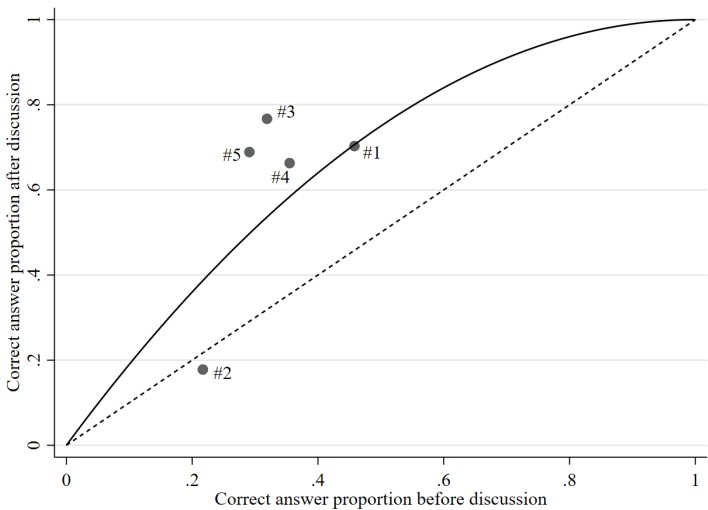

**Figure 1** **Scatter plot of correct answer proportion before and after discussion.** #1–#5: Measured value of ConcepTest #1–#5. Solid line: Theoretical value curve. Dotted line: Diagonal line, which means that there is no change in the correct answer proportion before and after the discussion.

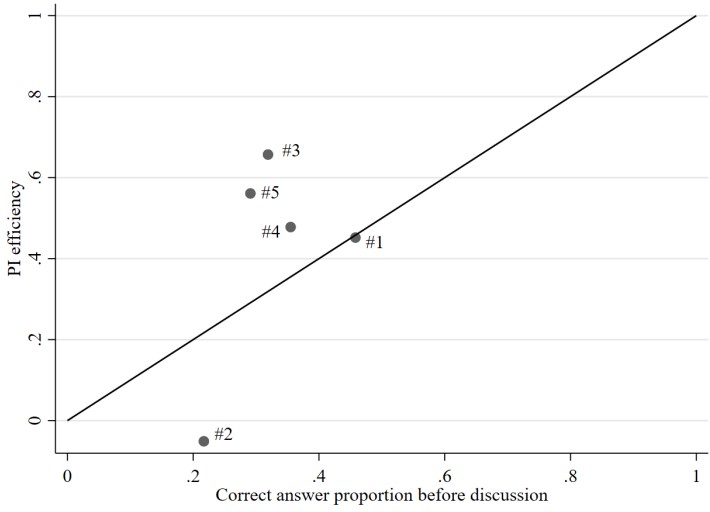

**Figure 2** **Scatter plot of correct answer proportion before discussion and PI efficiency.** #1–#5: Measured value of ConcepTest #1–#5. Solid line: Theoretical value curve.

## RESULTS

### Descriptive statistics

Table 1 provides an overview of the 121 students (34 women and 87 men) who enrolled for and attended the PI lecture. The numbers of respondents for each ConcepTest (before/after discussion) were 120/101 for #1, 115/107 for #2, 116/120 for #3, 107/101 for #4, and 110/106 for #5.
**Table 1  Number of students and respondents in each ConcepTest.**

| | | Number (percent) |
|---|---|---|
| Number of students | | 121 (100.0%) |
| Number of attendees | | 121 (100.0%) |
| Female | | 34 (28.1%) |
| Male | | 87 (71.9%) |
| Respondents of ConcepTest #1 | Before discussion | 120 (99.2%) |
| | After discussion | 101 (83.5%) |
| Respondents of ConcepTest #2 | Before discussion | 115 (95.0%) |
| | After discussion | 107 (88.4%) |
| Respondents of ConcepTest #3 | Before discussion | 116 (95.9%) |
| | After discussion | 120 (99.2%) |
| Respondents of ConcepTest #4 | Before discussion | 107 (88.4%) |
| | After discussion | 101 (83.5%) |
| Respondents of ConcepTest #5 | Before discussion | 110 (90.9%) |
| | After discussion | 106 (87.6%) |

**Table 2  Correct answer proportion and PI efficiency of ConcepTests #1~#5.**

| | #1 | #2 | #3 | #4 | #5 |
|---|---|---|---|---|---|
| Number of respondents before discussion | 120 | 115 | 116 | 107 | 110 |
| Number of respondents after discussion | 101 | 107 | 120 | 101 | 106 |
| Correct answer proportion before discussion ($N_b$) | 0.458 | 0.217 | 0.319 | 0.355 | 0.291 |
| Correct answer proportion after discussion ($N_a$) | 0.703 | 0.178 | 0.767 | 0.663 | 0.689 |
| Theoretical value of $N_a$ | 0.707 | 0.388 | 0.536 | 0.584 | 0.497 |
| PI efficiency ($\eta$) | 0.452 | −0.051 | 0.657 | 0.478 | 0.561 |
| Theoretical value of $\eta$ | 0.458 | 0.217 | 0.319 | 0.355 | 0.291 |
| Gap between measured and theoretical value | −0.007 | −0.268 | 0.338 | 0.123 | 0.270 |

## PI efficiency

Table 2 provides $N_b$, $N_a$, the theoretical value of $N_a$, $\eta$, the theoretical value of $\eta$, and the gap between measured $\eta$ and theoretical value of $\eta$. $N_b$ ranged from 0.217 to 0.458, and $N_a$ ranged from 0.178 to 0.767. Except in ConcepTest #2, $N_a$ was higher than $N_b$, and $\eta$ ranged from −0.051 to 0.657. In ConcepTest #2, $\eta$ was negative because the $N_a$ of 0.178 was less than the $N_b$ of 0.217. In this study, the average difference between the measured and theoretical values was 0.091, with a standard deviation of 0.216. A previous study on a physics PI lecture (*Nitta, Matsuura & Kudo, 2014*) revealed an average difference of 0.062 and a standard deviation of 0.219. Therefore, although the average difference in the present study is slightly higher, the overall results are similar. It was not possible in this study to examine gender differences in the answers, because gender data and ConcepTest answers were recorded separately on the student roster and Mentimeter, respectively, and were thus impossible to correlate.

To more intuitively grasp the relationship between the theoretical and measured values, Fig. 1 plots $N_b$ and $N_a$, and Fig. 2 plots $N_b$ and $\eta$. The solid lines in each figure represent the theoretical value, and the dotted line in Fig. 1 is a diagonal line, which indicates that the percentage of correct answers did not change after the discussion. In ConcepTest #1, the theoretical and measured values were almost the same; in ConcepTests #3 to #5, the measured value was higher than the theoretical value. On the other hand, in ConcepTest #2, the measured value was lower than the theoretical value, with a gap of −0.268 between the two. According to a previous study (Nitta, Matsuura & Kudo, 2014), a gap of lower than −0.2 was reported in about 14.3% of ConcepTests; therefore, the low gap in this study is not unusual.

## DISCUSSION

This study aimed to employ the ConcepTest to evaluate PI in a medical school epidemiology class. The goal was to measure the efficiency of active learning techniques and examine the usefulness of ConcepTests in determining effective active learning approaches. As education methods become increasingly student-centered, researchers and educators need to employ empirical research methods, such as well-defined measures, to determine their efficacy.

In this study, PI efficiencies varied widely, probably because efficiency depends on the field of study. For example, a mechanics lecture using a ConcepTest about position, velocity, and acceleration has high PI efficiency, whereas a ConcepTest on action and reaction has low PI efficiency (Kaneta & Nitta, 2009; Takahashi & Nitta, 2009; Nitta, Matsuura & Kudo, 2014). In epidemiology, the ConcepTest about the case-control study showed high PI efficiency ($\eta = 0.657$), but the ConcepTest about epidemic curves showed low PI efficiency ($\eta = -0.051$).

One of the factors that indicates the success of PI lectures is high PI efficiency (Nitta, 2010; Nitta, Matsuura & Kudo, 2014). Thus, a high learning effect might be obtained by presenting PI lectures in a field where the PI efficiency is high and conventional lectures in a field where the PI efficiency is low.

The proportion of correct answers before the discussion in this study ranged from 0.217 to 0.458, whereas the ideal range is said to be from 0.35 to 0.70 (Crouch & Mazur, 2001). In this lecture, the proportion of correct answers before the discussion was less than 0.35 in three ConcepTests. Thus, few students might have had a fruitful discussion. This effect was particularly large in ConcepTest #2 (0.217 proportion of correct answers before discussion), and the proportion of correct answers after the discussion was lower than that before the discussion. However, in other ConcepTests, the proportion of correct answers after the discussion was higher than the theoretical value even if the proportion of correct answers before the discussion was low. This suggests that student discussions may have a useful effect in learning epidemiology. Also, by using pre-learning activities, such as flipped classrooms, it may be possible to further increase PI efficiency (Rowley & Green, 2015; Zheng & Zhang, 2020; Sabale & Chowdary, 2020).

There are three reasons that low PI efficiency was demonstrated only in ConcepTest #2, which covered the epidemic curve. First, the lecture time may have been short. In the

other ConcepTests, only the concepts covered in the ConcepTest were explained before the test; however, in ConcepTest #2, an overall explanation of descriptive epidemiology was presented. As a result, little time was dedicated to explaining the epidemic curve. Second, the problem of false stereotypes may also have played a role. In ConcepTest #2, students were asked a question about the epidemic curve of food poisoning. Students may have believed the stereotype that food poisoning is caused mainly by bacteria. For food poisoning, it is necessary to carefully consider all causes when assessing the situation. However, many students may have considered only bacterial food poisoning, which may have resulted in the low PI efficiency. ConcepTest #2 was conducted with the pedagogical consideration that students should learn that it is necessary to think on the basis of epidemiological knowledge even for matters that are easily misunderstood, but it seemed to have a negative impact on PI efficiency. Third, the different phrasing of the questions in ConcepTest #2 may have affected the students' responses. The questions for the other four ConcepTests were phrased as ''choose the most appropriate option''. However, in ConcepTest #2, the question was phrased as ''choose the least appropriate option''. This may have caused confusion for students and affected PI efficiency.

This study has several limitations. First, since only one lecture and five ConcepTests were utilized in this study, it is difficult to generalize the results. The PI efficiency might depend on not only the difficulty of the ConcepTests but also factors such as the interactions among students (*Figueiredo & Figueiredo, 2020*), students' background knowledge (*Lasry, Mazur & Watkins, 2008*), and learning history (*Nitta, Matsuura & Kudo, 2014*), among others. These factors may have had a strong effect on PI efficiency. Therefore, based solely on the results of this study, it cannot be concluded that PI efficiency is low for ConcepTests regarding the epidemic curve and high for ConcepTests regarding the case-control study. Second, the evaluation was conducted only for each ConcepTest, not for the PI lecture as a whole. In physics, objective tests such as the Force Concept Inventory are used to evaluate entire PI lectures (*Hestenes, Wells & Swackhamer, 1992*). Further, there is a need for a subjective test for epidemiology. Third, the quality of the lectures may not be uniform. All lectures were conducted by the author, who provided explanations for all topics. Nonetheless, variations in quality can occur for each topic, and it should be noted that PI efficiency evaluates not only the ConcepTest, but a combination of the ConcepTest and pre-test lecture. Fourth, student discussions for each topic may not have been identical. While the discussion procedure was the same for all ConcepTests, students may not have been accustomed to engaging in discussions during class. Particularly at the beginning of the PI lecture, the effects of unfamiliarity with this format were considered significant. Therefore, in ConcepTests #1 and #2, the efficiency may have been lower than in the ConcepTests that followed. Furthermore, students in Asia, including Japan, were reported as not being active enough in active learning lectures (*Shimizu et al., 2019*). To avoid these situations, it was necessary to promote a smooth discussion by conducting an icebreaker before the PI lecture (*Martin & Bolliger, 2018*; *Basioudis, 2019*), or by incorporating e-learning into the lecture (*Shimizu et al., 2019*).

## CONCLUSIONS

This study reported the effectiveness of ConcepTests used in a PI lecture in epidemiology. Based on the results of five ConcepTests, this study showed that PI efficiency can be widely used in epidemiology lectures. This is one of the few studies that has tested PI efficiency in epidemiological education. Further, the differences in PI efficiencies between study topics were noteworthy. In some ConcepTests, even if the proportion of correct answers before the discussion was low, the proportion after the discussion was higher than the theoretical value. This suggests that PI lectures might be useful in epidemiology education.

However, the study had some limitations—mainly, a small number of ConcepTests and the difficulty of ensuring uniform quality across the PI lectures and discussions. Future research should address these issues, further utilizing ConcepTests in PI lectures and continually measuring PI efficiency. To evaluate the entire PI lecture, standardized tests are required, such as the Force Concept Inventory, which is an objective standardized test in mechanics (*Hestenes, Wells & Swackhamer, 1992*). However, as there is no widely accepted standardized test in epidemiology, it is necessary to develop objective standard tests.

## ACKNOWLEDGEMENTS

I am grateful to Yoko Oka for helping with the data management. I would like to thank Editage for English language editing.

### Funding

This study was supported by the Japan Medical Education Foundation. The funders had no role in study design, data collection and analysis, decision to publish, or preparation of the manuscript.

### Grant Disclosures

The following grant information was disclosed by the author:
Japan Medical Education Foundation.

### Competing Interests

The author declares there are no competing interests.

### Author Contributions

- Toshiharu Mitsuhashi conceived and designed the experiments, performed the experiments, analyzed the data, prepared figures and/or tables, authored or reviewed drafts of the paper, and approved the final draft.

### Human Ethics

The following information was supplied relating to ethical approvals (i.e., approving body and any reference numbers):

The study was approved by the Okayama University Graduate School of Medicine, Dentistry and Pharmaceutical Sciences and Okayama University Hospital, Ethics Committee (approval number K1909-037).

## Data Availability

The raw data and code are available in the Supplemental Files.

## Supplemental Information

Supplemental information for this article can be found online at http://dx.doi.org/10.7717/peerj.9640#supplemental-information.

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
