# Peer review of "Evaluation of epidemiological lectures using peer instruction: focusing on the importance of ConcepTests"

_PeerJ, doi:10.7717/peerj.9640_

## Round 0.1 · original submission · Major Revisions

Your manuscript was considered interesting and valuable, but one the reviewers raised a number of issues that need to be addressed and that will help improve your manuscript. Specifically, the reviewer suggests that you gather additional data in order to have a more complete study, including considering the impact of PI on students’ exam grades.

Please, submit a detailed rebuttal which shows where and how you have taken all comments and suggestions into consideration. If you do not agree with some of the reviewers’ comments or suggestions, please explain why. Your rebuttal will be critical in making a final decision on your manuscript. Please, note also that your revised version may enter a new round of review by the same or by different reviewers. Therefore, I cannot guarantee that your manuscript will eventually be accepted.

·

Basic reporting

Review of paper by Dr Mitsuhashi, Okayama University Hospital titled “Evaluation of epidemiologicval lectures using peer instructions. Focusing on the importance of concepTests.

The manuscript considers the effect of peer instructions conducted in a single 80 minutes lecture in epidemiology. Approx 120 students participated after providing written informed consent. Data consist of the outcome of 5 multiple choice questions before and after students were allowed to discuss the concept (Peer instructions, PI). In 3 cases outcome improved, in on case there was no difference, in the last of the 5 cases the outcome was worse than anticipated after PI.

Generally the manuscript is well written and easy to follow for the reader.

Abstract
The first-time-reader would benefit from having PI efficiency defined already in the abstract.

Introduction
Limited number of references provided,however all are relevant.
Few newer studies overlooked, e.g. Katyal et al, intl J Biomed and Adv Res 2016, reporting on among others the effect of PI in epidemiology lectures, and https://ucl.ac.uk/teaching-learning/case-studies/2018/sep/peer-instruction-transforms-medical-science-classroom.
There is quite a number of other new studies on PI in medical sciences as well, however not necessarily touching upon epidemiology.
I recommend the author to specify which gap this specific study addresses – which gap is filled. It is indirectly touched upon lines 62-66 & 71-72. Aim is clear, lines 73-74

Reference list: seems complete and without formatting errors
Tables and figures: reads well and seem of good resolution

Data tables and figures
2 excel files provided – one “appendix Table” with ConcepTests and “raw data responses” with conceptest outcomes before and after PI
Forms for informed consent + IRB approval enclosed.
Raw figures not encluded, but do.file is enclosed, which describes the code for generation of the graphs in stata

Experimental design

Methods section: described sufficiently and understandable.
Ethics are described. Approval obtained. It may be a matter of preference, but I recommend ethics section is moved up front in the methods section.
Author refers to “exercises and ConcepTests” line 82, were other activities than ConcepTests with PI used in the lectures?
How were students instructed to prepare for the lecture? As usual? Or were instructions different vs normal?

The data supporting the manuscript are limited! see section above.

The question is whether the manuscript in nature is more a case study, rather than a research study, since only one single lecture with 5 ConcepTests was used as data basis.

Suggestions are provided below to add data to provide the manuscript a more solid body.

Study / first data collection NOT confirmed by independent observations.

Specific suggestions for improving the data and discussion:

I find it difficult to understand which specific method (line 183) and empirical research methods (lines 184 and 185) the author is pointing at. Clarification /revision of sentence recommended.
In lines 186-195 the author discusses whether the topic taught will influence the efficiency of PI, did the author consider whether the way MCQs, i.e. The ConceptTests were phrased influenced the results? Were MCQ of the same type, although topic varied? Some variation is, as described – and as supported by references – expected. So 1 of 5 providing a worse outcome and 1 of 5 with no change seems quite good, but questions might require revision before being used another time. Any thought on this might be a valuable addition.
This links to the next questions, since the %of correct answers should be high enough (35% at least, as also referenced by Crouch and Mazur) to facilitate the PI. How was the nature of the most difficult ConcepTests in this study? What pedagogical considerations were behind the concepTests and design of concepTests used? see also comment below on design of MCQ.

Were any attempts done to prepare the students for the different lecture format before the lecture? Or were ConcepTests introduced “without any warning”?
Was culture considered as negatively influencing the students’ engagement in the PI? Cf Shimizu et al BMC Med Educ 2019
The study would benefit from including students’ evaluation of the use of concepTests, i.e.addition of qualitative data on students’ perception of the lectures. It is unclear whether this was considered in the limitations section (line225).
If students’ perception of the new teaching method cannot be added, did the author consider looking at the impact of the addition of PI on the exam grades? Maybe even comparing classes’ performances the year before and the current term (exam following october 2019) for exam questions in epidemiology? Some reports are available comparing students exam performance within the same course between lectures where PI was used and those taught conventionally cf. Carstensen et al Eur J Pharmacol. 2019

What was the teachers perception of using PI in epidemiology? Was it more difficult? Did it require much more time for preparing for the lecture? Was it difficult to use the mentimeter?

It is discussed that the negative outcome of ConcepTest 2 might be due to difficulties of the students to adhere or participate in the PI due to the novelty of the use of PI in the lectures. However, as the appendix Table indicate, the students are asked to choose the LEAST appropriate item. A good rule of thumb when preparing and designing MCQ is to not confuse students and to no ask for choices on negative terms like “choose the least appropriate” or “which of the following does NOT apply”. Cf e.g. Case and Swanson 2002 NBME, or Shank 2019 elearningIndustry. I recommend the author to add this point to the discussion for future investigators to consider design of the concepTests as well.

Validity of the findings

please see section above on suggestions for improvements.

The data body is very limited.
Limitations are discussed, few suggestions are indicated in section 2 for improvements.

underlying data are provided, nothing is wrong with these, however, more data might be available to support the study.

Given some of the references above (sections 1 and 2) I am afraid the author cannot conclude that this is the first report on PI efficency in epidemiological education (line 241), however the addition of the study is highly warranted cf Goldman et al Am J Epidem 2018

Additional comments

Author is commended on submitting data evaluating use of PI in the medical curriculum. Is a valuable contribution, however databody is very limited and author is recommended to consider whether further data can be added in support of the study cf sections 1-2-3 above.
language and structure is clear and manuscript is easy to read and follow

Reviewer 2 ·

Basic reporting

This article addresses and interesting and timely research topic, building on prior data outside of the field of medicine.

Professional English used, writing clear and concise.

Introduction contains a good overview of the subject material, major references cited. Good context for this study provided. Research question clearly stated.

Article structure appropriate, tables understandable and add value to text.

Article is self contiained

Experimental design

Experimental design is conventional, statistics are straightforward.
Methods described in sufficient detail to replicate
Appropriate ethical approval and oversight documented

Validity of the findings

Findings are valid and not overstated.
Findings are consistent with prior research in the subject area.
Underlying data provided, statistical analysis is reasonable

Conclusions based on the data, confined to the research question.

Reviewer 3 ·

Basic reporting

It is a good article that only describes what is significant about examining the effect of using Peer Instruction in epidemiological lectures. The article is written in clear English and with a structure that is easy to follow. In addition, the article contains relevant use of references and has a good introduction to the pedagogical method Peer Instruction. All appropriate raw data have been made available in in fine tables and associated graphs.

Experimental design

The article stays within the aims and scope of the journal.
The research question well defined and supported by relevant references. It is relevant and meaningful, and it is stated how the research fills the identified knowledge gap. The introduction to the pedagogical method Peer Instruction is clearly described. In the article referred to by Crouch & Mazur (2001), there is also a focus on pre-class activities such as reading and solving quizzes. These activities are important in relation to how well-prepared students worked with the ConceptTests in-class. So, in order to better understand your results, I suggest that you write a few sentences about what kinds of pre-class activities your students were asked to do.

The experimental design is well described and easy to follow and reproducible. The PI method is also well described. However, in step IV, it is unclear whether it is students with opposite answers to CT who should try to convince each other of what is the correct answer. Did your students move around to find other students with opposite answers or what exactly happened? There is a learning difference between whether the starting point of discussion is a cognitive conflict between two students or whether students discuss with someone with more or less the same opinion as themselves. Please add more details about how your students interacted.

Validity of the findings

The internal validity is good, although it is a small experiment. However, the external validity is not usable as the cohort effect is too small. The limitations of the study are well described and highlight very precisely which factors influence the results. In addition, the conclusions are well stated and linked to the original research question. Finally, the article highlights areas that need to be followed up with new studies to better determine the effect of using PI in epidemiological lectures.

Additional comments

You have written an interesting article regarding the use of PI in the field of epidemiological lectures. It could be interesting if you could experiment more with the preparatory online pre-class activities and at the same time investigate the in-class activities (PI lecture) in more detail. See, e.g. Rowley, N., & Green, J. (2015). Just-in-time teaching and peer instruction in the flipped classroom to enhance student learning. Education in Practice, 2 (1), 2015.

---

## Round 0.2 · accepted · Accept

You have done a thorough job of addressing the reviewers' comments, and as a result your manuscript is much improved. A minor point that I have is that there must be an error in line numbering. The last line number on page 3 is 69, but page 4 starts at line number 78. As a result, most of the line number references in your rebuttal letter were incorrect. This is a simple formatting error and I am sure it can be easily fixed.